# OpenReview forum: "Scalable Generative Modeling of Protein Ligand Trajectories via Graph Neural Diffusion Networks"
_ICLR.cc/2026/Conference — ICLR 2026 Conference Desk Rejected Submission_

### Official Review · Reviewer_ddcf · 2025-10-21

**Soundness:** 2
**Presentation:** 1
**Contribution:** 2
**Rating:** 2
**Confidence:** 5

**Summary:**

This paper presents HemePLM-Diffuse, a scalable generative framework for protein–ligand dynamics simulation that integrates SE(3)-invariant graph neural networks, time-aware cross-attention, and diffusion-based generative modeling.

The model represents proteins and ligands as bipartite SE(3)-equivariant graphs and learns to denoise molecular configurations through a diffusion process. A time-indexed cross-attention mechanism captures bidirectional interactions between protein residues and ligand atoms.

This architecture supports trajectory forecasting, transition path sampling, and ligand fragment inpainting, treating molecular dynamics as a “generative video” in 3D.

**Strengths:**

Combines geometric GNNs, temporal attention, and diffusion into a single differentiable framework for large-scale molecular trajectory generation.

The idea of unifying multiple downstream tasks, such as trajectory continuation, path sampling, and ligand fragment inpainting, is generally refreshing, though more details are needed for readers to fully understand them.

**Weaknesses:**

The main issue with the paper lies in its poor presentation quality, with inconsistent spacing between characters and symbols, duplicated related work sections, and missing technical details on the downstream applications (trajectory continuation, path sampling, and ligand fragment inpainting), making it difficult to follow.

Moreover, the related work discussion is incomplete: while the authors mention MDGen and AI2BMD, many other relevant deep learning–based approaches are omitted, resulting in an unbalanced and shallow literature review.

**Questions:**

NA

---

### Official Review · Reviewer_DH2H · 2025-10-29

**Soundness:** 1
**Presentation:** 1
**Contribution:** 1
**Rating:** 0
**Confidence:** 5

**Summary:**

The paper proposes HemePLM-Diffuse, a protein–ligand trajectory generator that combines an SE(3)-invariant GNN, a time-aware cross-attention module between protein and ligand nodes, and a diffusion process trained on MD trajectories. Claimed capabilities include forecasting trajectories, transition-path sampling, and ligand-fragment inpainting, with scale “beyond 10,000 atoms.” Results are summarized on a small collection (Ethanol, Alanine, Chignolin) and a HEME–protein (3CQV) case, with scatter plots of predicted vs. true energy/forces and brief baseline mentions (TorchMD-Net, MDGen, Uni-Mol).

**Strengths:**

The paper addresses the important and challenging problem of accelerating molecular dynamics simulations for large-scale protein-ligand systems. The high-level idea of combining equivariant GNNs with generative diffusion models is a relevant and active area of research for this problem.

**Weaknesses:**

The paper's entire premise is the "Generative Modeling of Protein Ligand Trajectories." However, the results section (Section 5) provides zero evidence of this capability.

All figures (Figs 2-5) and metrics (Sec 5.1, 5.6) are exclusively for static energy and force prediction, not dynamics or trajectories.

There are no trajectory plots, no RMSD-over-time analysis, no free energy surface comparisons, and no validation of transition paths.

The model's claimed ability to perform "trajectory forecasting" and "ligand fragment inpainting" (Sec 5.7) is asserted without any supporting data, plots, or metrics.

Claims of outperforming baselines like MDGen and Uni-Mol (Sec 5.8) are made without a single comparative metric or table.

The paper's claims on its key "scalability" contribution are completely incoherent.

The paper demonstrates a severe lack of care. Section 2 ("Related Work") and Section 3 ("Related Work") are identical, verbatim copies of each other. This is a disqualifying error on its own.

Contradictory and Poorly Reported Results: The few results that are presented are internally inconsistent.

**Questions:**

This paper requires a major overhaul. Given the fundamental issues, I ask the authors to sincerely revise their work for a future submission, as this manuscript is not ready for publication at this conference.

---

### Official Review · Reviewer_YJXo · 2025-10-30

**Soundness:** 1
**Presentation:** 1
**Contribution:** 1
**Rating:** 0
**Confidence:** 4

**Summary:**

The paper clearly points out the challenges molecular dynamics (MD) simulations face when dealing with large biomolecular systems and long biological timescales. It proposes a novel generative framework called HemePLM-Diffuse, which extends the MDGen architecture to handle larger and more complex biomolecular systems. The model achieves promising performance in predicting the system’s energies and force fields.

**Strengths:**

1. The motivation is  rational.
2. The work extends the MDGen architecture to handle larger and more complex biomolecular systems.

**Weaknesses:**

1. The paper claims to surpass leading methods, including TorchMD-Net, MDGen, and Uni-Mol, in trajectory prediction and transition path sampling. However, it does not provide clear or quantitative comparative results to substantiate this claim.
2. The paper has significant writing issues. Section 4 is logically disorganized, and the first two paragraphs of Section 4.1 are highly repetitive. The core contribution of the model is not clearly emphasized, and the authors fail to properly explain how the energy and force estimations are obtained.
3. The numerical results reported in Sections 5.1 and 5.6 are completely inconsistent, making it impossible to assess the validity of the model.
4. The paper repeatedly emphasizes that the proposed model is specifically designed for large-scale systems and can be extended to systems with more than 10,000 atoms. However, all the quantitative and visual results are based on very small molecules. There is no evidence in the manuscript to support its central claim of scalability.

**Questions:**

The same as Weaknesses.

---

### Official Review · Reviewer_TuVt · 2025-10-31

**Soundness:** 1
**Presentation:** 1
**Contribution:** 1
**Rating:** 0
**Confidence:** 5

**Summary:**

The paper purports to describe a generative model for protein-ligand trajectories. What is actually presented are minimal results on an energy/force prediction discriminative model.

**Strengths:**

Scalable generative modeling of protein ligand trajectories is an important problem.

**Weaknesses:**

The claims of this paper are not supported by the results, many of which are absent or underspecified.  It seems that what was trained was a model to predict energies and forces (Fig 1).  Minimal results (10 ligands) are provided.  Training data, model architecture, and training algorithms are all underspecified.  There are issues with the flow and writing of the paper (e.g. the related work section is duplicated).

This work is not ready for publication.

**Questions:**

How does the model described in Figure 1 achieve the stated goals of the paper of performing trajectory generation?  What is the training data? How is the Pearson correlation between predicted and reference trajectories calculated?

---

### Note · Program_Chairs · 2026-01-17
**Submission Desk Rejected by Program Chairs**

The following references in this submission do not refer to real documents and/or have major errors in bibliographic information:

 E. Brini, G. Jayachandran, and M. Karplus. Coarse-graining biomolecular simulations via statistical learning. J. Chem. Phys., 154:040901, 2021.
João M. L. Ribeiro et al. End-to-end learning of molecular force fields using differentiable molecular mechanics. Journal of Chemical Physics, 153(12):124109, 2020.